# INTERNAL-COORDINATE DENSITY MODELLING OF PROTEIN STRUCTURE: COVARIANCE MATTERS

## ABSTRACT

After the recent ground-breaking advances in protein structure prediction, one of the remaining challenges in protein machine learning is to reliably predict distributions of structural states. Parametric models of fluctuations are difficult to fit due to complex covariance structures between degrees of freedom in the protein chain, often causing models to either violate local or global structural constraints. In this paper, we present a new strategy for modelling protein densities in internal coordinates, which uses constraints in 3D space to induce covariance structure between the internal degrees of freedom. We illustrate the potential of the procedure by constructing a variational autoencoder with full covariance output induced by the constraints implied by the conditional mean in 3D, and demonstrate that our approach makes it possible to scale density models of internal coordinates to full protein backbones in two settings: 1) a unimodal setting for proteins exhibiting small fluctuations and limited amounts of available data, and 2) a multimodal setting for larger conformational changes in a high data regime.

## 1 INTRODUCTION

Proteins are macro-molecules that are involved in nearly all cellular processes. Most proteins adopt a compact 3D structure, also referred to as the *native state*. This structure is a rich source of knowledge about the protein, since it provides information about how the protein can engage biochemically with other proteins to conduct its function. The machine learning community has made spectacular progress in recent years on the prediction of the native state from the amino acid sequence of a protein (Jumper et al., 2021; Senior et al., 2020; Wu et al., 2022b; Baek et al., 2021; Wu et al., 2022a). However, the static picture of the structure of a protein is misleading: in reality a protein is continuously moving, experiencing both thermal fluctuations and larger conformational changes, both of which affect its function. One of the remaining challenges in machine learning for structural biology is to reliably predict these distributions of states, rather than just the most probable state. We discuss the state of the density modelling field in Section 5 (Related work).

Modelling the probability density of protein structure is non-trivial, due to the strong constraints imposed by the molecular topology. The specific challenges depend on the chosen structural representation: if a structure is represented by the 3D coordinates of all its atoms, these atom positions cannot be sampled independently without violating the physical constraints of e.g. the bond lengths separating the atoms. In addition, an arbitrary decision must be made about how the structure is placed in a global coordinate system, which implies that operations done on this representation should preferably be invariant or equivariant to this choice. An alternative is to parameterize the structure using internal coordinates, i.e. in terms of bond lengths, bond angles and dihedrals (rotations around the bonds). The advantage of this representation is that internal degrees of freedom can be sampled independently without violating the local bond constraints of the molecule. It also makes it possible to reduce the number of degrees of freedom to be sampled – for instance fixing the bond lengths to ideal values, since they fluctuate much less than the torsion angles and bond angles.

For the reasons given above, an internal coordinate representation would appear to be an attractive choice for density modelling. However, one important problem reduces the appeal: small fluctuations in internal coordinates will propagate down the chain, leading to large fluctuations remotely downstream in the protein. As a consequence, internal-coordinate density modelling necessitates careful modelling of the covariance structure between the degrees of freedom in order to ensure that small

fluctuations in internal coordinates result in small perturbations of the 3D coordinates of the protein. Such covariance structures are typically highly complex, making direct estimation difficult.

In this paper, we investigate whether density modelling of full-size protein backbones in internal coordinates is feasible. We empirically demonstrate the difficulty in estimating the covariance structure of internal coordinates from data, and instead propose a technique for *inducing* the covariance structure by imposing constraints on downstream atom movement using the Lagrange formalism. Rather than estimating the covariance structure from scratch, we can instead modulate the covariance structure by choosing appropriate values for allowed fluctuations of downstream atoms. We demonstrate the procedure in the context of a variational autoencoder (Fig. 1).

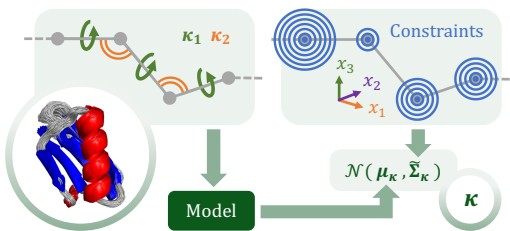

Figure 1: A protein structure ensemble is modelled in internal coordinate space, while imposing constraints on atom fluctuations in Euclidean space. The resulting full covariance structure can be used to sample from a multivariate normal distribution.

Given a prior on the internal coordinate fluctuations and a predicted mean, we impose constraints on the atom fluctuations in 3D space to obtain a full covariance structure over the internal coordinates. We show that this allows us to generate valid structures in terms of both internal and Cartesian coordinates. Our method is validated in two regimes: a low data regime for proteins that exhibit small, unimodal fluctuations, and a high data regime for proteins that exhibit multimodal behavior. We anticipate that this method could serve as a building block applicable more generally for internal-coordinate density estimation, for instance internal-coordinate denoising diffusion models.

**Our main contributions are:**

- We formulate a procedure for inducing full protein backbone covariance structure in internal coordinates, based on constraints on atom fluctuations in 3D space.
- Rather than predicting a full covariance matrix over internal coordinates, our proposed method only requires to predict one Lagrange multiplier for each atom, from which the full covariance matrix can be constructed. For $M$ atoms, this corresponds to a reduction from $(2 \times M - 5)^2$ to simply $M$ predicted values.
- We design a variational autoencoder which models fluctuations for full-length protein backbones in internal coordinates. Even though constraints are formulated in Euclidean space, the model is not dependent on a global reference frame (i.e. it is rotationally invariant).
- We demonstrate that our model provides meaningful density estimates on ensemble data for proteins obtained from experiment and simulation.

**Scope.** The focus of this paper will be on modelling distributions of protein structure states in internal coordinates. We are thus concerned with thermodynamic ensembles, rather than the detailed dynamics that a molecule undergoes. Dynamics could potentially be modelled on top of our approach, for instance by fitting a discrete Markov model to describe transitions between states, and using our approach to model the thermal fluctuations within a state, but this is beyond the scope of the current work. Another perspective on our approach is that we wish to describe the aleatoric uncertainty associated with a static structure.

## 2 BACKGROUND

### 2.1 CARTESIAN VS INTERNAL COORDINATES

As stated before, Cartesian coordinates and internal coordinates each have advantages and disadvantages. Assume we have a 3D protein structure in Euclidean space with atom positions $x$. Throughout this paper, we only consider backbone atoms N, $C_\alpha$ and C, which means that the total number of atoms $M = 3 \times L$, with $L$ the number of amino acids. The Euclidean setting thus results in $3 \times M$ coordinates. Even though in this setting each of the atoms can fluctuate without affecting other atoms in the backbone chain, there is no guarantee for chemical integrity, i.e. conservation of bond lengths

and respecting van der Waals forces. This can lead to backbone crossings and generally unphysical protein structures.

One way to ensure chemical integrity is to parameterize protein structure in internal coordinate space using dihedrals $\boldsymbol{\kappa_1}$, bond angles $\boldsymbol{\kappa_2}$ and bond lengths $\boldsymbol{\kappa_3}$. Here, dihedrals are torsional angles that twist the protein around the bond between two consecutive atoms, bond angles are angles within the plane that is formed by two consecutive bonds, and bond lengths are the distances between two consecutive backbone atoms. Since bond length distributions have very little variance, we choose to fix them, thereby reducing the number of variables over which we need to estimate the covariance. We will refer to the remaining two internal coordinates together as $\boldsymbol{\kappa}$ to avoid notation clutter. As dihedrals are defined by four points (the dihedral is the angle between the plane defined by the first three points and the plane defined by the last three points) and bond angles are defined by three points, the resulting protein structure representation will have $(2 \times M) - 5$ coordinates. Not only does this result in less coordinates to determine a full covariance structure over, the coordinates are also automatically rotation and translation invariant, as opposed to Cartesian coordinates.

The remaining problem is that small changes in one internal coordinate can have large consequences for the global structure of the protein, since all atoms downstream of the internal coordinate will move together, acting like a rigid body. It is therefore challenging to preserve global structure while altering internal coordinates, since they are mostly descriptive of local structure.

## 2.2 STANDARD PRECISION ESTIMATORS DO NOT CAPTURE GLOBAL FLUCTUATIONS

Because of the limitations of internal coordinates mentioned in Section 2.1, it is a highly non-trivial task to capture a full covariance structure over $\boldsymbol{\kappa}$ which also conforms to constraints in Euclidean space that are inherent to the protein. As an example, we use a standard estimator to get a precision matrix (i.e. the inverse of the covariance matrix) over $\boldsymbol{\kappa}$ for a short molecular dynamics (MD) simulation on "1pga", also known as "protein G" ( Fig. 2). Details about the simulation can be found in Appendix A. We see that when we take samples from a multivariate Gaussian over $\boldsymbol{\kappa}$ with the true mean (based on the dataset) and the estimated precision, the samples exhibit atom fluctuations that are much higher than the original simulation, and with a very different pattern.

To overcome the limitations that regular covariance and precision estimators have, we incorporate constraints on atom fluctuations in Euclidean space.

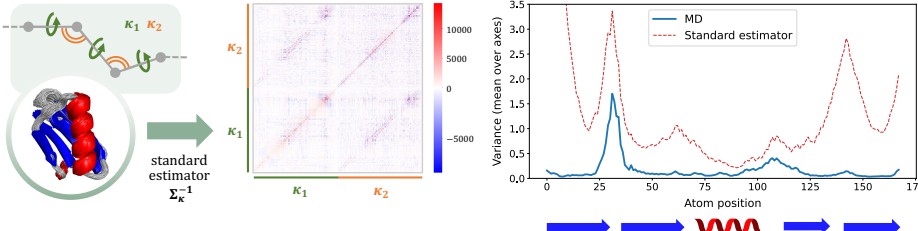

Figure 2: When a standard estimator is used to get the precision structure over internal coordinates, resulting atom fluctuations significantly deviate from MD simulations. Blue arrows and red helices represent secondary structural elements. The variance is calculated as the mean of the variances over the x, y and z axis, in $\mathring{A}^2$.

## 3 INTERNAL-COORDINATE DENSITY MODELLING WITH CONSTRAINTS

### 3.1 SETUP

We parameterize a 3D protein structure in terms of internal coordinates (i.e. dihedrals and bond angles, while bond lengths are kept fixed), which together will be referred to as $\boldsymbol{\kappa}$. Our aim is to obtain a multivariate Gaussian distribution over the deviations from the mean $p(\Delta\boldsymbol{\kappa})$, centered at zero, with a full precision structure. This target distribution is subject to constraints over atom fluctuations, enforcing the preservation of global structure. We have a prior $q(\Delta\boldsymbol{\kappa})$, which we will call the $\kappa$-prior, over the internal coordinate distribution, where the mean is zero and the precision is a diagonal matrix with the diagonal filled by the inverse variance over all $\boldsymbol{\kappa}$ values $\sigma_{\boldsymbol{\kappa},\text{data}}^{-2}$, estimated from our input

data. The strength of the $\kappa$-prior can be tuned using hyperparameter $a$. The $\kappa$-prior is defined as

$$q(\Delta\boldsymbol{\kappa}) = \frac{1}{Z_q} \exp\left(-\frac{1}{2}\Delta\boldsymbol{\kappa}^{\mathrm{T}}\boldsymbol{\Sigma}_{\boldsymbol{\kappa},\mathrm{prior}}^{-1}\Delta\boldsymbol{\kappa}\right), \tag{1}$$

where $Z_q = \boldsymbol{\sigma}_{\boldsymbol{\kappa},\mathrm{data}}\sqrt{2\pi a}$ is the normalization constant for the $\kappa$-prior distribution and $\boldsymbol{\Sigma}_{\boldsymbol{\kappa},\mathrm{prior}}^{-1} = a \cdot \mathrm{diag}(\sigma_{\boldsymbol{\kappa},\mathrm{data}}^{-2})$. Our approach will be to construct a new distribution $p$ which is as close as possible to $q$, but which fulfills a constraint that prohibits the downstream 3D coordinates from fluctuating too much. We thus wish to minimize the Kullback-Leibler divergence between the objective distribution and $\kappa$-prior:

$$\mathcal{D}_{\mathrm{KL}}(p|q) = \int p(\Delta\boldsymbol{\kappa})\ln\frac{p(\Delta\boldsymbol{\kappa})}{q(\Delta\boldsymbol{\kappa})}\mathrm{d}\Delta\boldsymbol{\kappa}, \tag{2}$$

adding constraints on the expected value over squared atom displacements of each downstream atom:

$$\mathbb{E}_{\Delta\boldsymbol{\kappa}\sim p(\Delta\boldsymbol{\kappa})}\left[\Delta x_m^2\right] = C_m \tag{3}$$

where $\mathbb{E}_{\Delta\boldsymbol{\kappa}\sim p(\Delta\boldsymbol{\kappa})}\left[\Delta x_m^2\right]$ is the expected value for the squared displacement of atom $m$, and $C_m$ is a constant equivalent to the variance of the atom position $\sigma_{x_m}^2$ assuming equal variance in all directions (isotropic Gaussian). Since every $\Delta x_m$ is a function of $\Delta\boldsymbol{\kappa}$ with probability density function $p(\Delta\boldsymbol{\kappa})$, we can use the law of the unconscious statistician to reformulate the constraints as follows:

$$\mathbb{E}_{\Delta\boldsymbol{\kappa}\sim p(\Delta\boldsymbol{\kappa})}\left[\Delta x_m^2\right] = \int \Delta x_m^2 p(\Delta\boldsymbol{\kappa})\mathrm{d}\Delta\boldsymbol{\kappa} = C_m \tag{4}$$

## 3.2 LAGRANGE FORMALISM TO INCORPORATE CONSTRAINTS

Employing Jaynes' maximum entropy principle (Jaynes, 1957), we use the Lagrange formalism to incorporate $M$ of these constraints, with $M$ the number of atoms, under the conditions that our probability density $p(\Delta\boldsymbol{\kappa})$ has zero mean and sums to one[1]. This leads to Lagrangian

$$\tilde{\mathcal{D}}(p,q) = \int p(\Delta\boldsymbol{\kappa})\ln\frac{p(\Delta\boldsymbol{\kappa})}{q(\Delta\boldsymbol{\kappa})}\mathrm{d}\Delta\boldsymbol{\kappa} + \lambda_0\left(\int p(\Delta\boldsymbol{\kappa})\mathrm{d}\Delta\boldsymbol{\kappa} - 1\right)$$
$$+ \sum_{m=1}^{M}\lambda_m\left(\int \Delta x_m^2 p(\Delta\boldsymbol{\kappa})\mathrm{d}\Delta\boldsymbol{\kappa} - C_m\right). \tag{5}$$

Next, we take the functional derivative and set it to zero: $\frac{\partial\tilde{\mathcal{D}}(p,q)}{\partial p(\Delta\boldsymbol{\kappa})} = 0$, leading to the following well-established result (Jaynes, 1957; Kesavan & Kapur, 1989)[2]:

$$0 = \ln\frac{p(\Delta\boldsymbol{\kappa})}{q(\Delta\boldsymbol{\kappa})} + 1 + \lambda_0 + \sum_{m=1}^{M}\lambda_m\Delta x_m^2 \Rightarrow p(\Delta\boldsymbol{\kappa}) = \frac{1}{Z_p}q(\Delta\boldsymbol{\kappa})\exp\left(-\sum_{m=1}^{M}\lambda_m\Delta x_m^2\right) \tag{6}$$

with $Z_p = \exp(-1-\lambda_0)$ the normalization constant of the target distribution. Note that $\frac{\partial^2\tilde{\mathcal{D}}(p,q)}{\partial p(\Delta\boldsymbol{\kappa})^2} = \frac{1}{p(\Delta\boldsymbol{\kappa})}$ is positive, therefore we know our solution will indeed be a minimum.

## 3.3 FIRST ORDER APPROXIMATION FOR ATOM FLUCTUATIONS

In order to use Eq. (6) to satisfy the imposed constraints, we need to express $\Delta x^2$ in terms of $\Delta\boldsymbol{\kappa}$. To first order, we can express the displacement vectors $\Delta\boldsymbol{x}_m$ of each atom as a regular small angle approximation (first order Taylor expansion):

$$\Delta\boldsymbol{x}_m \approx \sum_i \frac{\partial\boldsymbol{x}_m}{\partial\kappa_i}\Delta\kappa_i \tag{7}$$

---

[1]Even though throughout this derivation we have included the normalization constant for rigor, in practice we work with unnormalized densities and normalize post hoc, since we recognize the final result to be Gaussian.

[2]Note that $\frac{\partial}{\partial y}\ln y/q + \lambda_0(y-1) + \sum_{m=1}^{M}\lambda_m\left(\Delta x_m^2 y - C_m\right) = \ln\frac{y}{q} + y\cdot\frac{1}{y} + \lambda_0 + \sum_{m=1}^{M}\lambda_m\Delta x_m^2$

where $\boldsymbol{x}_m$ is the position of the $m^{\text{th}}$ atom, under the condition that the atom is *post-rotational* (Bottaro et al., 2012), i.e., the location of atom $m$ is downstream of the $i^{\text{th}}$ internal coordinate. From Eq. (7) it follows that the squared distance can be approximated by

$$\Delta x_m^2 \approx \left\| \left( \sum_i \frac{\partial \boldsymbol{x}_m}{\partial \kappa_i} \Delta \kappa_i \right)^2 \right\| = \sum_{ij} \left( \frac{\partial \boldsymbol{x}_m}{\partial \kappa_i} \Delta \kappa_i \cdot \frac{\partial \boldsymbol{x}_m}{\partial \kappa_j} \Delta \kappa_j \right) = \Delta \boldsymbol{\kappa}^{\mathrm{T}} \boldsymbol{G}_m \Delta \boldsymbol{\kappa} \,, \qquad (8)$$

where $\boldsymbol{G}_m^{i,j} = \frac{\partial \boldsymbol{x}_m}{\partial \kappa_i} \cdot \frac{\partial \boldsymbol{x}_m}{\partial \kappa_j}$ is a symmetric and positive-definite matrix.

Substituting Eq. (8) and our $\kappa$-prior expression from Eq. (1) into our target distribution from Eq. (6) gives a new Gaussian distribution:

$$p(\Delta \boldsymbol{\kappa}) \approx \frac{1}{\tilde{Z}} \exp \left( -\frac{1}{2} \Delta \boldsymbol{\kappa}^{\mathrm{T}} \left( \boldsymbol{\Sigma}_{\boldsymbol{\kappa},\mathrm{prior}}^{-1} + \boldsymbol{\Sigma}_{\boldsymbol{\kappa},\mathrm{constr}}^{-1} \right) \Delta \boldsymbol{\kappa} \right) = \mathcal{N}(0, \tilde{\boldsymbol{\Sigma}}_{\boldsymbol{\kappa}}) \,,$$

where $\tilde{Z}$ is the new normalization constant, $\boldsymbol{\Sigma}_{\boldsymbol{\kappa},\mathrm{constr}}^{-1} = 2 \sum_{m=1}^{M} \lambda_m \boldsymbol{G}_m$ and the covariance matrix of the new Gaussian distribution $\tilde{\boldsymbol{\Sigma}}_{\boldsymbol{\kappa}} = \left( \boldsymbol{\Sigma}_{\boldsymbol{\kappa},\mathrm{prior}}^{-1} + \boldsymbol{\Sigma}_{\boldsymbol{\kappa},\mathrm{constr}}^{-1} \right)^{-1}$.

### 3.4 SATISFYING THE CONSTRAINTS

The final step in the constrained optimization is to establish the values for the Lagrange multipliers. A closed form solution for this is not readily available, but using the findings from Section 3.3, we can now rewrite the constraints from Eq. (4) as

$$C_m = \mathbb{E}_{\Delta \boldsymbol{\kappa} \sim \mathcal{N}(0, \tilde{\boldsymbol{\Sigma}}_{\boldsymbol{\kappa}})} \left[ \Delta \boldsymbol{\kappa}^{\mathrm{T}} \boldsymbol{G}_m \Delta \boldsymbol{\kappa} \right] = \mathrm{tr}(\tilde{\boldsymbol{\Sigma}}_{\boldsymbol{\kappa}} \boldsymbol{G}_m) \qquad (9)$$

where the last simplification step comes from standard expectation calculus on a quadratic form ($\Delta \boldsymbol{\kappa}^{\mathrm{T}} \boldsymbol{G}_m \Delta \boldsymbol{\kappa}$), where $\Delta \boldsymbol{\kappa}$ has zero mean (Eq. 378 in Petersen et al. (2008)). Although it is nontrivial to express Lagrange multipliers $\lambda$ in terms of atom fluctuations $C$, we thus see that it is possible to evaluate $C$ given a set of Lagrange multipliers $\lambda$. In the following, we will therefore construct our models such that our networks predict $\lambda$, directly.

### 3.5 VAE PIPELINE

**VAE model architecture.** To demonstrate how our method works within a modelling context, we use a variational autoencoder (VAE), as illustrated in Fig. 3. The VAE has a simple linear encoder that takes internal coordinates $\boldsymbol{\kappa}$ (dihedrals and bond angles, bond lengths are kept fixed) as input and maps to latent space $\boldsymbol{z}$, where we have a standard Gaussian as a prior on the latent space, which we call $z$-prior to avoid confusion with the $\kappa$-prior. The decoder outputs the mean over $\boldsymbol{\kappa}$, which is converted into Cartesian coordinates using pNeRF (AlQuraishi, 2018). This mean structure in 3D coordinates is used for two purposes. First, using the structure we can evaluate the partial derivatives of atom positions with respect to the individual $\kappa$ as in Eq. (7). Second, the predicted mean over $\boldsymbol{\kappa}$ is used to get a pairwise distance matrix $\boldsymbol{d}$ that serves as the input to a U-Net (Ronneberger et al., 2015), from which we estimate values for the Lagrange multipliers for each constraint. This allows the variational autoencoder, conditioned on the latent state z, to modulate the allowed fluctuations. Implementation-wise, the U-net is concluded with an average pooling operation that for each row-column combination computes one Lagrange multiplier $\lambda$. Together with our fixed-variance $\kappa$-prior over $\boldsymbol{\kappa}$ and hyperparameter $a$ determining the strength of this $\kappa$-prior, a new precision matrix is formed according to Eq. (9). The model can generate new structures through simple ancestral sampling: first generating $\boldsymbol{z}$ from the standard normal $z$-prior, and subsequently sampling from a multivariate Gaussian distribution with the decoded mean and the constructed precision matrix. For specific model settings see Appendix A.

**Loss.** We customarily optimize the evidence lower bound (ELBO) using the Gaussian likelihood on the internal degrees of freedom as constructed above. This likelihood does not ensure that the predicted Lagrange multipliers are within the range within which our first order approximation of the fluctuations is valid. To ensure this, we add an auxiliary regularizing loss in the form of a mean absolute error over $\lambda^{-1}$, which prevents the $\kappa$-prior from dominating. By tuning the weight $w_{\mathrm{aux}}$ on the auxiliary loss, we can influence the strength of the constraints.

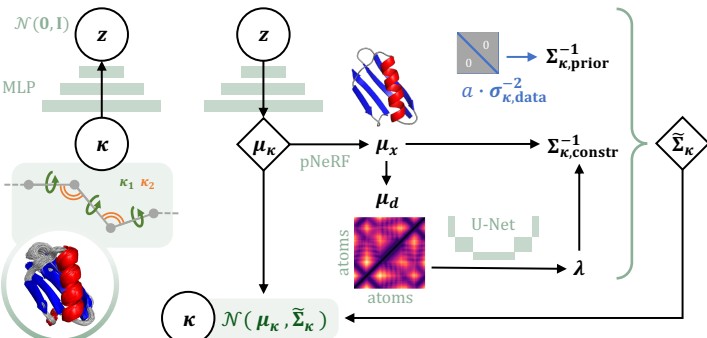

Figure 3: Model overview. The encoder (left) embeds internal coordinates into the latent space. The decoder (right) predicts a mean, from which constraints are extracted to obtain a precision matrix. Together with the $\kappa$-prior over the precision matrix based on the input data, a new precision matrix is formed which can be used to sample from a multivariate Gaussian.

# 4 EXPERIMENTS

## 4.1 TEST CASES

**Unimodal setting in low data regime.** We consider three test proteins for small fluctuations in a low data regime: 1unc, 1fsd, and 1pga. 1unc corresponds to the solution structure of the human villin C-terminal headpiece subdomain. This protein contains 36 residues, corresponding to 108 backbone (N, $C_\alpha$ and C) atoms. This solution nuclear magnetic resonance (NMR) dataset is freely available from the Protein Data Bank and contains 25 conformers. 1fsd, a beta beta alpha (BBA) motif, is also a freely available NMR dataset containing 41 structures. This system has 28 residues with 84 backbone atoms. 1pga, corresponding to B1 immunoglobulin-binding domain protein G, is a 56 amino acid long protein with 168 backbone atoms. We have a short in-house molecular dynamics (MD) simulation, which is 20ns long and structures were saved at a 50ps interval, resulting in 400 structures for this protein. See Appendix A for more details about the simulation.

**Multimodal setting in high data regime.** We also include two test cases for larger fluctuations following a multimodal distribution in a high data regime. Both are known as "fast-folders" and the MD datasets were obtained from Lindorff-Larsen et al. (2011). We refer the reader to this work for detailed descriptions of the simulations. Chignolin (cln025) is a peptide with a hairpin structure, containing 10 residues and thus 30 backbone atoms. The simulation is 106 μs long, saved at a 200 ps interval, resulting in 534.743 data points. The second test case, 2f4k, is the chicken villin headpiece, with 35 residues and 105 backbone atoms. The simulated trajectory is 125 μs and also saved every 200 ps, yielding 629.907 structures.

## 4.2 METRICS

For the unimodal setting, we choose two simple measures of local and global structure, respectively. To evaluate local structure fluctuations, we show Ramachandran plots, a well-known visualization tool in the context of protein structures, where $\phi$ and $\psi$ dihedrals, which are the torsional angles around the $N - C_\alpha$ and $C_\alpha - C$ bonds, are plotted against each other. As a global measure, we report the variance over atom positions, averaged over three dimensions, across superposed (i.e. structurally aligned) samples to evaluate global structure fluctuations.

For the multimodal setting, we report free energy landscapes, parameterized by the first two components of time-lagged independent component analysis (TICA) (Molgedey & Schuster, 1994). Similar to e.g. PCA, TICA fits a linear model to map a high-dimensional input to a lower-dimensional output, but TICA also incorporates the time axis. The resulting components are ranked according to their capacity to explain the slowest modes of motion. Taking the first two components corresponds to selecting reaction coordinates that underlie the slowest protein conformational changes, which is highly correlated with (un)folding behavior. We fit the TICA model on the time-ordered MD data, and pass samples from the VAE and baselines through the fitted model to create free energy landscapes.

**Baselines.**   Apart from comparing the generated samples from our model to reference distributions from MD or NMR, we include four baselines. The first baseline, named "$\kappa$-prior (fixed)" is a VAE trained to predict $/mu_\kappa$ given a fixed covariance matrix that is equal to $\Sigma_{\kappa,\mathrm{prior}}^{-1}$. In other words, this is the same as our full VAE setup, but omitting the imposed 3D constraints. The second baseline is "$\kappa$-prior (learned)", which corresponds to a more standard VAE-setting where the decoder directly outputs a mean and a variance (i.e. a diagonal covariance matrix). The third baseline does not involve a VAE, but samples structures from a multivariate Gaussian with a mean based on the dataset and a precision matrix computed by a standard estimator. This is an emperical estimator for MD datasets, and an Oracle Approximating Shrinkage (OAS) estimator (Chen et al., 2010) for NMR datasets, since emperical estimators do not converge for such low amounts of samples. Finally, we include the flow-based model from Köhler et al. (2023) as a fourth baseline, which, to the best of our knowledge, is the current state of the art in density modelling for internal coordinate representations.

## 4.3   INTERNAL-COORDINATE DENSITY MODELLING RESULTS

**Unimodal setting in low data regime.**   The unimodal, low data regime test cases exhibit small fluctuations around the native protein structure, where the largest fluctuations correspond to the loops connecting different secondary structure elements. Fig. 4 demonstrates that 1unc, 1fsd and 1pga structures sampled from the VAE conform to local and global constraints, with valid Ramachandran distributions when compared to the reference as well as improved atom position variance along the chain compared to the baselines (see quantitative results in Table A2). Even in the extremely low data regime of 25 and 41 data points for 1unc and 1fsd, respectively (top two rows in Fig. 4), the VAE is able to estimate a full covariance matrix that approximates the distribution better than the baselines, especially in loop regions. This can lead to unphysical structures with backbone crossings, even though the local structure is preserved. These effects can also be observed in the 3D visualization of the sampled ensembles in Fig. A2.

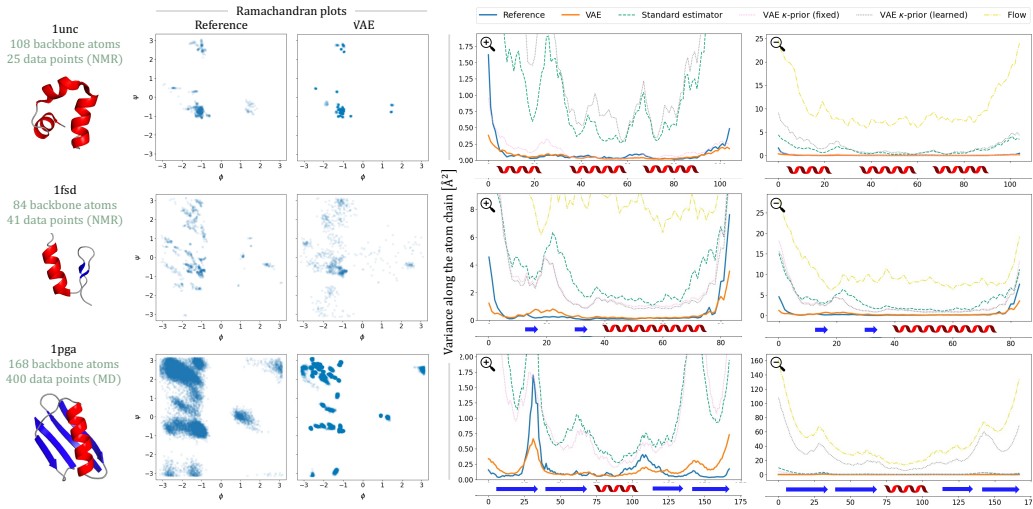

Figure 4: Modelling fluctuations in the unimodal setting for 1pga, 1fsd, and 1unc. Left: structure visualization, with α-helices in red and β-sheets as blue arrows. Middle: Ramachandran plots for the MD reference and VAE samples. Right: variance along the atom chain for VAE samples, MD reference, and baselines. Secondary structure elements are indicated along the x-axis.

The third test case, 1pga (bottom row in Fig. 4), has a more complex structure with two β-strands at the N-terminus forming a sheet together with two β-strands from the C-terminus. These global constraints are not captured well by the baselines in this low data regime, resulting in very high fluctuations in loop regions which violate the native structure (additional visualizations can be found in Fig. A2). For our VAE, we see the benefits of imposing global constraints, resulting in much better density estimation compared to the baselines. Moreover, we can control the interplay between local and global constraints by adjusting the hyperparameters of our model, as exemplified in Appendix D.1. However, the complexity of this protein prevents perfect density estimation in a low data regime.

Interestingly, we show in Appendix C.1 that the variance of the atom positions highly correlates to the imposed constraints $C$ calculated from a set of predicted Lagrange multipliers using Eq. (9).

**Multimodal setting in high data regime.** Here, we explore the use of our approach for modelling more complex behavior in a high data regime. Fig. 5 shows the free energy landscape in terms of the two first TICA components for cln025 and 2f4k. When comparing the VAE and the baselines to the MD reference (see also quantitative results in Table A3), it is clear that the learned prior and standard estimator do not capture all modes in the free energy landscape. The flow-based model performs best, suggesting that in this multimodal setting with plenty of available data, our proof-of-concept VAE is not as expressive as this state-of-the-art model. Moreover, the benefit of imposing 3D constraints on top of the fixed $\kappa$-prior (see baseline) seems beneficial for cln025, but the effect is not as strong for the $\alpha$-helical 2f4k, where local constraints might dominate (a similar effect can be seen when comparing the VAE to the fixed $\kappa$-prior baseline for 1unc in the unimodal setting). However, our simple VAE setup is evidently able to model large conformational changes through its latent space, demonstrating how our general-purpose method for modeling fluctuations using 3D constraints can be incorporated into more expressive models to model complex behavior.

Similar to the unimodal case, there is a tradeoff between local and global constraints which we can modulate using hyperparameters, as demonstrated in Appendix D.2. In addition, we show in Appendix C.3 how distinct regions of the VAE latent space map to different clusters in the TICA free energy landscape, and visualize the corresponding structures.

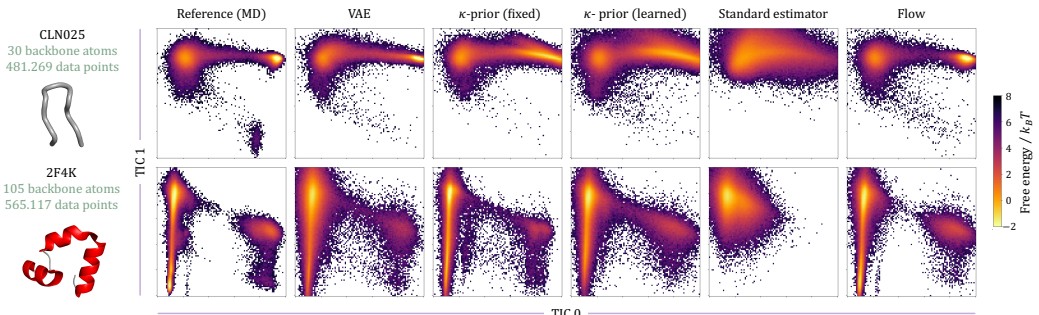

Figure 5: Modelling (un)folding behavior in the multimodal setting for cln025 and 2f4k. Left: structure visualization. Right: TICA free energy landscapes for MD reference, VAE, and baselines.

## 5 RELATED WORK

There is a large body of work on models for analyzing trajectories of molecular dynamics simulations, either through Markov state models (Chodera & Noé, 2014; Singhal & Pande, 2005; Sarich et al., 2013; Schütte et al., 1999; Prinz et al., 2011), or more complex modelling strategies (Mardt et al., 2018; Hernández et al., 2018; Sultan et al., 2018; Mardt et al., 2020; Xie et al., 2019). Typically, these focused on dimensionality reduced representations of the molecular structures, and are therefore not density models from which samples can be drawn.

To our knowledge, the first generative density model of full protein coordinates was the Boltzmann generator (Noé et al., 2019), a normalizing flow over the Cartesian coordinates of protein ensembles. An extension of this approach was later used to estimate $C_\alpha$-only coarse-grained force fields for molecular dynamics simulations (Köhler et al., 2023). This method, which uses internal coordinate inputs, demonstrated the ability of augmented flows to sample structural ensembles for proteins up to 35 amino acids . Other approaches involve latent variable models. One example is the IG-VAE, which generates structures in 3D coordinates but expresses the loss in terms of distances and internal coordinates to maintain SE(3) invariance. Similar approaches have been used to analyze cryo-EM data, where the task is to generate ensembles of structures given the observed cryo-EM image data. Since cryo-EM data provides information at slightly lower resolution than the full-atomic detail we discuss here, the output of these approaches are often density maps in 3D space (Zhong et al., 2021; Punjani & Fleet, 2021). One example of atomic-level modelling in this space is Rosenbaum et al. (2021), which decodes deterministically into 3D coordinates, but describes the variance in

image space. Finally, diffusion models have recently provided a promising new approach to density modelling, with impressive examples of density modelling at the scale of full-size proteins (Watson et al., 2022; Ingraham et al., 2022; Anand & Achim, 2022).

The primary objective in our paper is to investigate whether internal-coordinate density modelling with a full covariance structure is feasible using a simple, parsimonious setup. Internal-coordinate probabilistic models of proteins have traditionally focused on protein local structure, i.e. correct modelling of angular distributions of the secondary structure elements in proteins. Early work was based on hidden Markov models of small fragments (Camproux et al., 1999; 2004; de Brevern et al., 2000; Benros et al., 2006). The discrete nature of the fragments meant that these models did not constitute a complete probabilistic model of the protein structure. Later models solved this issue by modelling local structure in internal coordinates, using different sequential models and angular distributions (Edgoose et al., 1998; Bystroff et al., 2000; Hamelryck et al., 2006; Boomsma et al., 2008; 2014; Thygesen et al., 2021). Due to the downstream effects of small internal-coordinate fluctuations, these models are not by themselves capable of modelling the distribution of entire protein structures, but they are useful as proposal distributions in Markov chain Monte Carlo (MCMC) simulations of proteins (Irbäck & Mohanty, 2006; Boomsma et al., 2013). Using deep learning architectures to model the sequential dependencies in the protein chain, recent work has pushed the maximum length of fragments that can be reliably modelled to length 15 (Thygesen et al., 2021), where the fragment size is limited due to the challenges in estimating the necessary covariance structure.

Our work was inspired by methods used for constrained Gaussian updates in MCMC simulation, first introduced by Favrin et al. (2001), and later extended by Bottaro et al. (2012). Our method generalizes the approach to global updates of proteins, derives the relationship between the Lagrange multipliers and corresponding fluctuations in Euclidean space, and uses neural networks to govern the level of fluctuations in order to modulate the induced covariance structures.

Recent work has demonstrated that internal-coordinate modelling can also be done using diffusion models (Jing et al., 2022). So far this method has been demonstrated only on small molecules. We believe the method we introduce in this paper might help scale these diffusion approaches to full proteins. In Cartesian space, the Chroma model (Ingraham et al., 2022) demonstrated the benefits of correlated diffusion arising from simple constraints between atoms. Our method can be viewed as an extension of this idea to richer covariance structures.

# 6 DISCUSSION

Although protein structure prediction is now considered a solved problem, fitting the density of structural ensembles remains an active field of research. Many recent activities in the field focus on diffusion models in the Cartesian coordinate representation of a protein. In this paper, we take a different approach, and investigate how we can describe small-scale fluctuations in terms of a distribution over the internal degrees of freedom of a protein. The main challenge in this context is the complex covariance between different parts of the chain. Failing to model this properly results in models that produce disruptive changes to the global structure even for fairly minor fluctuations in the internal coordinates. Instead of estimating the covariance matrix from data, we show that it can be induced by imposing constraints on the Cartesian fluctuations. In a sense, this represents a natural compromise between internal and Cartesian coordinates: we obtain samples that are guaranteed to fulfill the physical constraints of the local protein topology (e.g. bond lengths, and bond angles), while at the same time producing meaningful fluctuations globally.

We implement the idea in the decoder of a variational autoencoder on two protein systems. This is primarily a proof of concept, and this implementation has several limitations. First of all, the standard deviations in the $\kappa$-prior of the internal degrees of freedom are currently set as a hyperparameter. These could be estimated from data, either directly within the current VAE setup, or using a preexisting model of protein local structure. Another limitation is the current model is that the produced fluctuations are generally too small to fully cover the individual modes of the target densities. This could be solved by constructing a hierarchical VAE, where samples are constructed as a multi-step process, similar to the generation process in diffusion models. In fact, we believe that our fundamental approach of induced covariance matrices could be a fruitful way to make diffusion models in internal coordinates scale to larger systems, by allowing for larger non-disruptive steps. We leave these extensions for future work.

## 7 CODE AND DATA AVAILABILITY

Code and in-house generated data will be made available upon acceptance.

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

## A  EXPERIMENTAL DETAILS

**Model.**  See Fig. 3 for an overview of the model. The encoder and the decoder of the VAE are simple three-layer MLPs (multilayer perceptrons). Given a protein with $M$ backbone atoms, the encoder takes $(2 \times M - 5) \times 2$ inputs, corresponding to $(M - 3)$ dihedrals and $(M - 2)$ bond angles which are fed in as pairs of $(\sin, \cos)$ inputs to avoid periodicity issues. Similarly, the decoder yields $(2 \times M - 5) \times 2$ outputs, which can be converted to angles in the $[-\pi, \pi]$ interval using the 2-argument arctangent (`atan2`). The MLP linear layer sizes of the encoder are $[128, 64, 32]$, mapping to a 16-dimensional latent space, and layer sizes of the decoder are $[32, 64, 128]$ (reverse of the encoder).

We use a standard U-Net (Ronneberger et al., 2015) to predict atom fluctuation constraints $\lambda$ from the mean predictions that the decoder outputs. The predicted mean for the internal degrees of freedom $\boldsymbol{\mu_\kappa}$ is translated into a mean structure $\boldsymbol{\mu_x}$ using pNeRF (AlQuraishi, 2018), from which we calculate a pairwise distance matrix. This $M \times M$ matrix with a single "channel" serves as the input to the U-Net, which scales the number of channels up to $1024$ in four steps before scaling back down to one channel in four steps.

All datasets were split 90%-10% into a training and validation set. The best model is selected based on the validation loss. The weights for the $\kappa$-prior and auxiliary loss were explored with grid search (see Appendix D), values chosen for the models reported in the main paper are shown in Table A1 together with other experimental details. The model training starts with a warm-up phase in two different ways: 1) predicting $\boldsymbol{\mu_\kappa}$ only, with $\boldsymbol{\Sigma} = \mathbf{I}$ and 2) linearly increasing the weight of the KL-term from 0 to 1. Proteins in the low data regime (unimodal setting) have a 100 epoch mean-only warm-up and a 200 epoch KL warm-up, while proteins in the high data regime (multimodal setting) have a 3 epoch mean-only warm-up and an 8 epoch KL warm-up. All models were trained using an Adam optimizer with a learning rate of $5\mathrm{e}^{-4}$, on a Nvidia Quadro RTX (48GB) GPU.

Final metrics are calculated on structures sampled from the model. For the evaluation in the unimodal setting, the number of samples was chosen to be equal to the total number of data points (25, 41 and 400 for 1unc, 1pga and 1fsd, respectively). For the multimodal cases, 400.000 samples were drawn for TIC analysis. TICA was done using the `Deeptime` library (Hoffmann et al., 2021), using a lagtime of 100 and reducing the high-dimensional input to two dimensions. The TICA model is fit on the reference data (ordered in time), from which the resulting linear map is stored and applied to sampled structures from the VAE and baselines. All structure visualizations were done using PyMOL (Schrödinger, version 2.5.2).

Table A1: Experimental details for test cases.

|  | # train | # validation | # residues | # epochs | batch size | a | $\mathbf{w}_{\mathrm{aux}}$ |
|---|---|---|---|---|---|---|---|
| **1unc** | 23 | 2 | 36 | 1000 | 32 | 50 | 1 |
| **1fsd** | 37 | 4 | 28 | 1000 | 32 | 25 | 1 |
| **1pga** | 360 | 40 | 56 | 1000 | 32 | 50 | 25 |
| **cln025** | 481269 | 53474 | 10 | 50 | 64 | 25 | 50 |
| **2f4k** | 565117 | 62790 | 35 | 50 | 32 | 50 | 1 |

**Molecular dynamics details.**  The molecular dynamics simulation for 1pga was done in OpenMM (Eastman et al., 2017), using an Amber forcefield (Maier et al., 2015), water type TIP3P, box geometry "rhombic dodecahedron" and a padding of 1 nm on each side of the solvated protein (i.e. 2 nm in total). The simulation is 20ns in total with a 50ps time lag, giving 400 structures. For MD details on cln025 and 2f4k we refer the reader to Lindorff-Larsen et al. (2011).

# B QUANTITATIVE RESULTS

## B.1 UNIMODAL SETTING, LOW DATA REGIME

Table A2: MSE (lower is better) to reference for atom fluctuations, unimodal setting.

|  | VAE | $\kappa$-prior (fixed) | $\kappa$-prior (learned) | Standard estimator | Flow |
|---|---|---|---|---|---|
| **1unc** | 0.021 | 2.080 | **0.013** | 5.888 | 122.490 |
| **1fsd** | **0.585** | 13.949 | 12.732 | 9.666 | 107.052 |
| **1pga** | **0.040** | 3.654 | 1.709 | 1154.914 | 3157.693 |

## B.2 MULTIMODAL SETTING, HIGH DATA REGIME

Table A3: Jensen-Shannon distance (lower is better) between binned Boltzmann distributions, i.e. $\exp\left(-\frac{\text{free energy}}{k_B T}\right)$, comparing VAE and baselines to the reference, multimodal setting.

|  | VAE | $\kappa$-prior (fixed) | $\kappa$-prior (learned) | Standard estimator | Flow |
|---|---|---|---|---|---|
| **cln025** | 0.456 | 0.539 | 0.606 | 0.686 | **0.194** |
| **2f4k** | 0.373 | 0.297 | 0.342 | 0.517 | **0.183** |

## C  VAE SAMPLING

### C.1  COMPARING CONSTRAINTS TO ATOM FLUCTUATIONS ACROSS SAMPLES

As derived in Eq. (9), we can evaluate the constraint value $C_m$ for each atom $m$ given a set of Lagrange multipliers. These constraints were placed on the squared atom displacements, which is equivalent to the variance along the atom chain. Fig. A1 demonstrates that the isotropic fluctuations of 400 1pga samples drawn from the VAE are indeed quite close to $C$ calculated from 400 separately sampled sets of Lagrange multipliers. Since the constraints are placed on non-superposed (i.e. not structurally aligned) protein structures[3], this plot shows the variance along the atom chain for non-superposed structures.

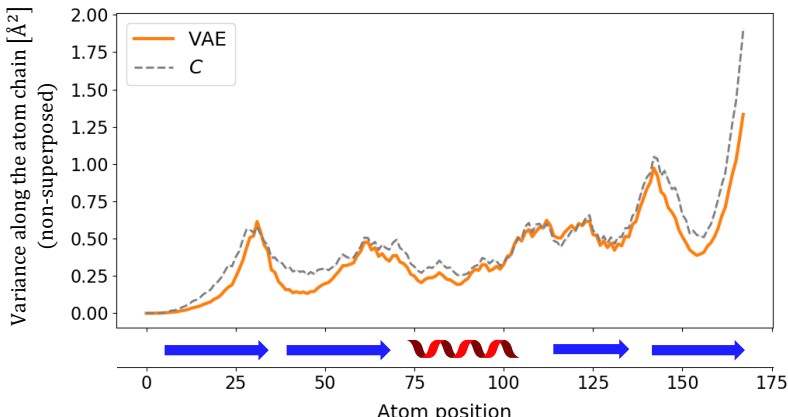

Figure A1: Variance along the atom chain for non-superposed 1pga structures sampled with the VAE (orange) compared to constraints $C$ calculated from predicted $\lambda$-values (grey dashed). Secondary structure element locations are indicated.

### C.2  VISUALIZATION OF SAMPLED STRUCTURES IN THE UNIMODAL SETTING

Fig. A2 shows sampled superposed ensembles for our model and baselines, as well as the MD/NMR reference. This demonstrates that VAE samples, where global constraints were enforced, generally have globally consistent fluctuations compared to the reference data. In contrast, the baselines tend to exhibit fluctuations that are too large, which can lead to unphysical structures containing crossings and, in some cases, lacking secondary structure elements.

---

[3]Sampled protein structures are built using pNeRF(AlQuraishi, 2018), which builds the chain step-by-step, thereby corresponding to our post-rotational constraints.

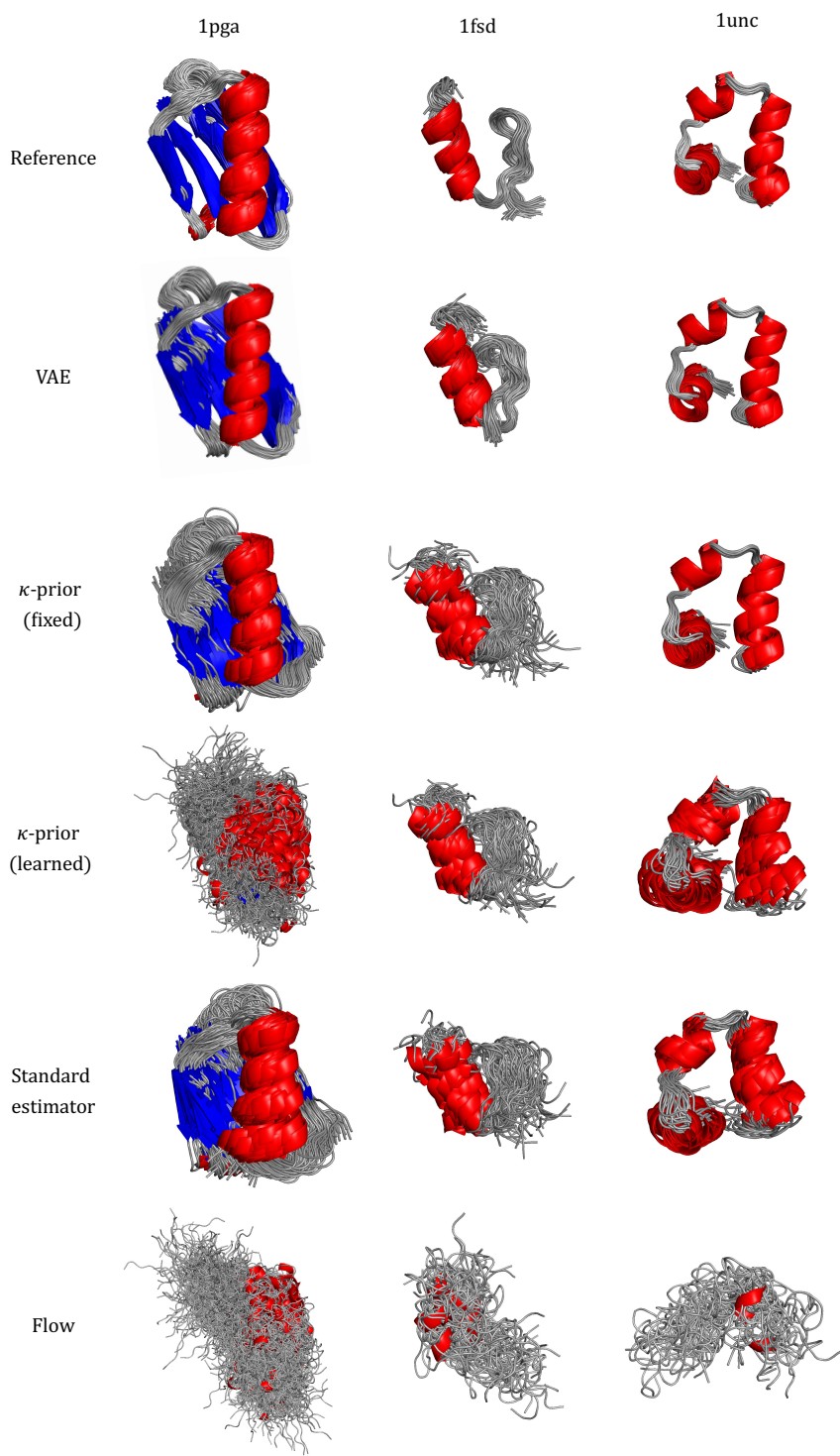

Figure A2: Visualization of ensembles for reference data, the VAE model and baselines for 1pga, 1fsd and 1unc. Number of samples is equal to the reference ensemble (400, 41 and 25 for 1pga, 1fsd and 1unc, respectively.)

## C.3 LATENT SPACE VISUALIZATION IN THE MULTIMODAL SETTING

In this section, we visualize the VAE latent space in the multimodal setting (cln025) in Fig. A3. Moreover, we demonstrate how 100 random samples from latent space map to structure samples in the TICA free energy landscape, and show the 3D structures that correspond to these samples. Transitions from the native state to more unfolded conformations can be observed when going from the cluster in the top right of TICA space towards the left. Depending on $\tilde{\Sigma}$, fluctuations around the means (which are decoded from the latent space samples) can vary in size. Therefore, means that are close together in terms of latent space location do not necessarily lead to sampling similar 3D structures. Moreover, we used a UMAP to reduce the number of latent dimensions from 16 to 2, and this simplified representation might not capture the full complexity of the latent space. Nonetheless, it is apparent that more unfolded structures largely originate from the rightmost cluster in latent space.

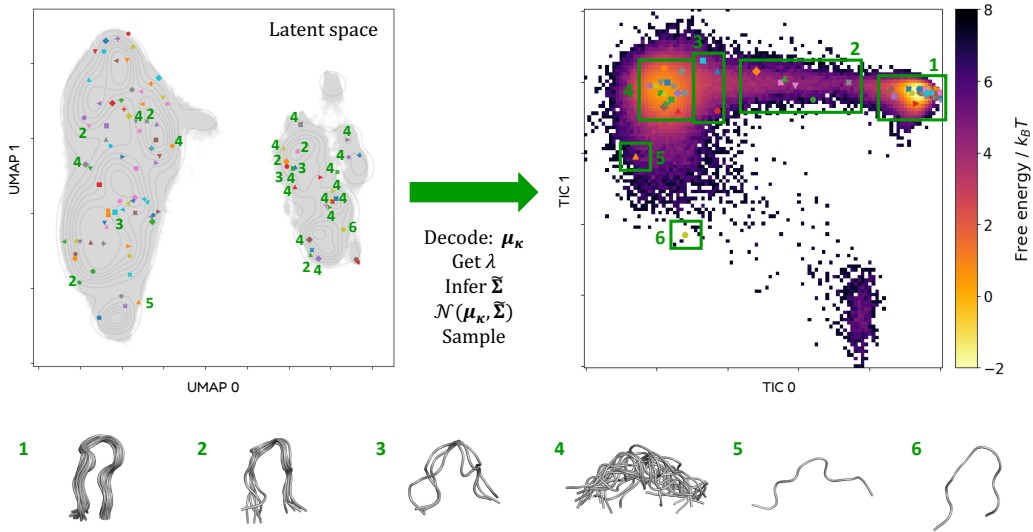

Figure A3: Top left: UMAP reduction to 2D of the originally 16-dimensional VAE latent space, with a 100 samples shown in random shapes and colors. The grey scatterplot depicts the aggregated posterior, with the KDE of the aggregated posterior as grey lines. Annotated green numbers correspond to boxes in the TICA free energy landscape (all structures corresponding to box 1 are left unlabelled to avoid clutter). Top right: structure samples corresponding to latent space samples visualized in TICA space with the same symbols and colors as the latent space samples. Samples are grouped together in green numbered boxes. Bottom row: 3D structures corresponding to the different numbered boxes in the TICA plot.

## D    ABLATION FOR HYPERPARAMETERS

The two main hyperparameters that need to be chosen in the VAE setting are the strength of the $\kappa$-prior $a$, and the weight of the regularizing loss $w_{\text{aux}}$. These two weights can be set to prioritize local or global constraints in different ways. We demonstrate the effect on a unimodal case (protein G, 1pga) and a multimodal case (chignolin, cln025). In both cases, results are shown for a gridsearch over $a = [1, 25, 50]$ and $w_{\text{aux}} = [1, 25, 50]$.

### D.1    UNIMODAL

Fig. A4 shows results for the ablation on $a$ and $w_{\text{aux}}$ in the unimodal setting. Increasing the strength of the $\kappa$-prior through $a$ while keeping $w_{\text{aux}}$ constant corresponds to narrower distributions in the Ramachandran plot and bond angle distributions. A higher weight $w_{\text{aux}}$ for a constant $a$ leads to stronger global constraints, as demonstrated by the fluctuations along the atom chain.

### D.2    MULTIMODAL

To understand the impact of hyperparameters in the multimodal setting, we first consider the impact on samples drawn from the VAE that was trained with a fixed $\kappa$-prior, which depends on hyperparameter $a$. Fig. A5 illustrates how the distribution in the TIC free energy landscape changes when strengthening the prior. For $a = 1$, there is a preference towards the metastable cluster on the top left, while increasing the value of $a$ leads to a stronger preference for the lowest energy cluster on the top right.

When sampling from the VAE, where constraints are imposed on top of the $\kappa$-prior, there is interplay between $a$ and $w_{\text{aux}}$, as shown in Fig. A6. Even though the exact trend is less clear here, the relative values of the hyperparameters have an observable influence on e.g. the width of the "bridge" between the topmost two clusters, the size of the higher-energy downward extrusion of the top left cluster, and the spread towards the less populated cluster on the bottom right.

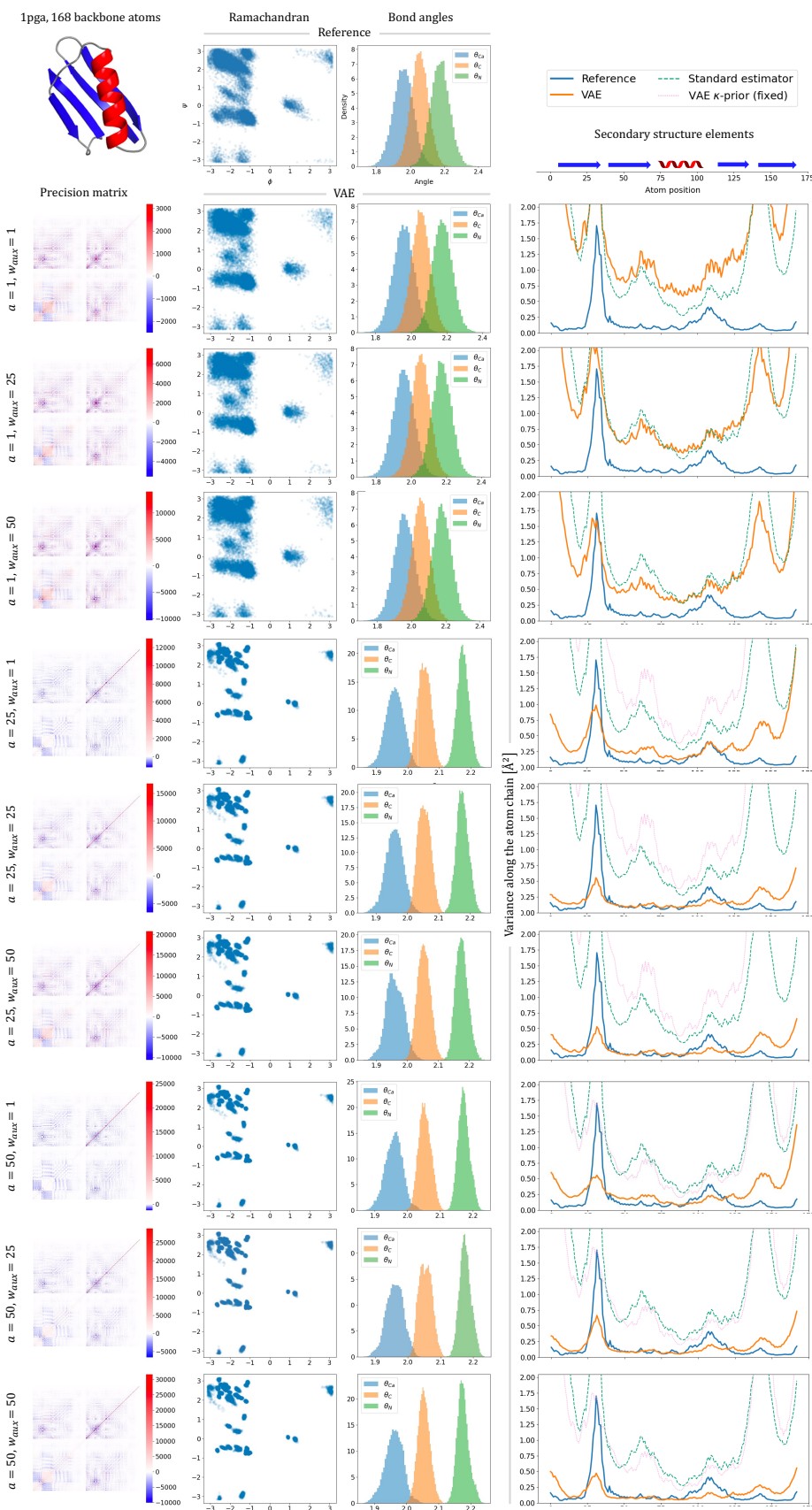

Figure A4: Ablation of $a$ and $w_{\mathrm{aux}}$ for protein G (1pga, structure shown at top left). From left to right: precision matrix example predicted by the VAE, Ramachandran plot, bond angle distributions, fluctuations along the atom chain (secondary structure elements indicated, VAE $\kappa$-prior (fixed) out of scale for $a = 1$).

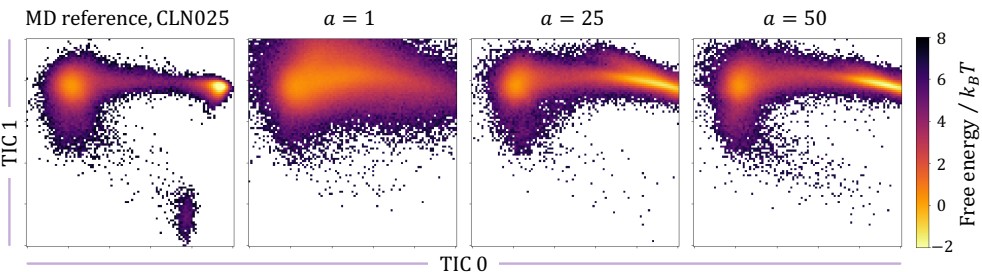

Figure A5: Influence of hyperparameter $a$ on samples drawn from the VAE with a fixed $\kappa$-prior (without imposing constraints) for chignolin (cln025), visualized in TIC space.

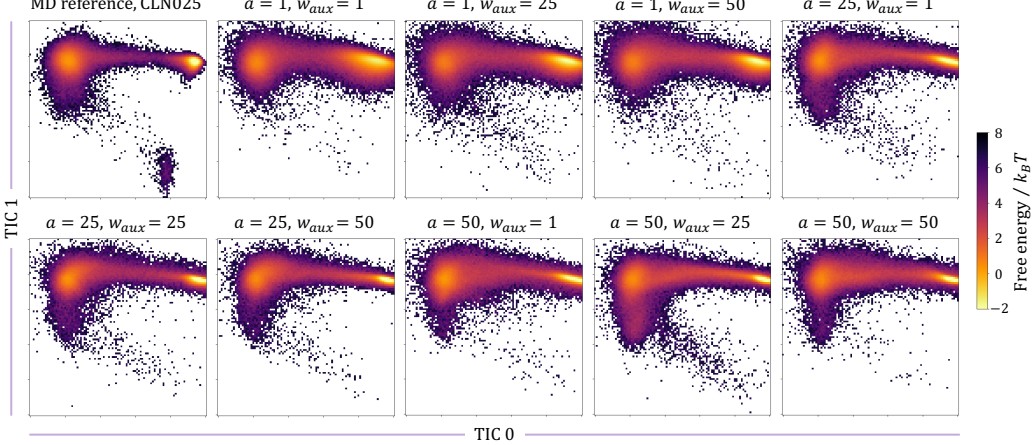

Figure A6: Hyperparameter ablation of $a$ and $w_{\mathrm{aux}}$ VAE samples for chignolin (cln025), visualized in TIC space.

