# OpenReview forum: "Internal-Coordinate Density Modelling of Protein Structure: Covariance Matters"
_ICLR.cc/2024/Conference — Submitted to ICLR 2024_

### Official Review · Reviewer_U1Uk · 2023-10-30

**Soundness:** 1 poor
**Presentation:** 2 fair
**Contribution:** 3 good
**Rating:** 6
**Confidence:** 3

**Summary:**

This paper presents a generative model for protein angles such that they try to not change absolute coordinates too much. The paper takes a functional Lagrangian perspective.

--

Post-response update: The mathematical concerns were adressed during discussion. I think this is a good and clever contribution to the field, while still having some weakness in demonstrating its ML significance.

**Strengths:**

- The main idea of deriving how angles and coordinates are coupled through Lagrangian is outstanding. The paper is well written and method is well derived until around half-way. I found this paper super interesting to read, and this is definitely a promising approach.
- The results show relatively good generation of variations wrt reference, but there are still quite a lot of room to improve.

**Weaknesses:**

- The math in this paper is sloppy and incomplete, and I’m not convinced that the method is correctly derived. I suspect that the Lagrangian is incorrectly derived, but I would be happy to see full derivations of the method to verify if it is. The results have some issues, which implies that something is perhaps wrong.
- I could not understand the generative model with VAEs and Unets, or the experiments. I’m not sure what the paper or the models try to demonstrate or achieve, despite the clear problem statement of finding safe distributions of angles wrt coordinates. I think the paper deviates from the original problem quite far into somewhere else. I can’t really interpret if the results are good or not.
- The ML significance of the method is ultimately a bit light. There are only a few experiments, and no comparisons to other ML models. It seems that this paper is about finding variations in proteins. These are conformers, and there is already ML works on doing generative models for those. Comparisons to those are needed, and the paper should better positions itself to wider literature. Currently the paper takes too narrow and isolated perspective to the problem domain.

**Questions:**

- Why do you need to include the integral=1 constraint? Isn’t p automatically a proper distribution?
- I find the notation to be a bit odd. The eq 5 should be defined as a lagrangian; it's not a KL anymore. It has sufficient and necessary conditions for minima, and these should be properly formulated and presented. I’m not sure what is a “functional derivative of KL wrt p(Delta x)”. What is a functional derivative? How can you take a derivative wrt a density evaluation at one point? Which point? All of them? Only one? Can you present this concept in more detail? There is an adhoc flavour to this, and I suspect something that the rigor is insufficient to actually claim you are doing a Lagrangian. You need to rigorously derive and present the Lagrangian formalism. If you want to take a functional viewpoint, then you need to derive the functional representations of the objects, eg. a hilbert representation of the distributions.
- Why does the \int p d term vanish in the derivative? Why does p(deltak) vanish as well? Surely this should not happen: Shouldn't you need to apply product rule and get Dp * log p/q + p * D(logp/q), where “D” denoted derivatives. I am not convinced this is correct. Please include all intermediate steps so we can verify its correct. Please include rigorous math that shows all the derivatives properly.
- Similarly, I have trouble understanding eq 7. What does the approx do? How good of an approximation this is? Where does this come from? Please include all intermediate derivations, properly introduce and expose your math, and please include citations to textbooks or works where similar approaches have been done earlier.
- How does an i’th angle relate to m’th atom i’th coordinate? How come you have the same i'th index for two different things (phi/phi and x/y/z)?
- Why is there no i in eq 8 suddenly? What is i and j? Why do we take a dot product? I have trouble following the story.
- If you always normalise all Gaussians, why did you include the int=1 constraint?
- I’ve lost track where eq 9 comes from. What is this? Aren't Cm user defined constants: why are we now solving for their values? Also aren’t you solving for the lambdas? Why are there none here? It seems that again quite a lot of intermediate derivations are missing, and I can’t verify that this is correct.
- What does it mean that you predict lambdas with a network? Err… Lagrangian multipliers can’t be just chosen at will or predicted: the lagrangian formalism dictates under which coefficient values you can say something about your solution optimality. If you don’t follow the lagrangian approach rigorously, you have no guarantees of what you end up with, and it might be of little use, or not do what you think it does. Are you perhaps not even doing a lagrange method but instead perhaps doing a penalty method? Can you clarify the setting and the goals and the overall workflow here?
- I’m lost on what are you trying to achieve with the VAE, and what’s the point. So you take angles, encode them into latents, and reconstruct the angles. Why is this useful? What are we trying to achieve? If the reason you have a VAE is to evaluate derivatives, then something is wrong. Surely you should be able to just compute your derivatives directly?
- So the VAE reconstruction as fed to a Unet that outputs lagrange multipliers… This feels adhoc. Why is it useful to predict lagrange multipliers? Intuitively this sounds wrong: the lagrange multipliers need to be at equilibrium point under lagrange conditions (etc), and not just predicted by some neural network. Can you elaborate what are you trying to do, and what role the lagrange has here? How much rigor are you even aiming at?
- It seems that in the end you only produce a kind of soft regularisation by something that is inspired or slightly flavored with Lagrangians, but you are not actually doing Lagrangian optimisation. Given that the section 3 is named “constraints” I was under the impression that you are actually doing Lagrangian constraints, but I don’t think this is the case. Can you clarify what role the Lagrangian has?
- I’m quite confused what is the overall pipeline, and why you are doing what you are doing. I thought that you wanted to include constraints such that varying angles doesn't change the absolute coordinates too much. But in fig 3 the C’s are not even present anymore, and it seems that the constraints now come dynamically from a Unet, based on coordinate means. What if the coordinate variances are large? What guarantees the Unet to give sensible constraints? How do we even define sensible constraints? What does the final N(mu,tildeSigma) represent, or try to do? Does this distribution still have something to do with constraints? What does “weighed by Lagrange multipliers” mean? Surely you can’t just multiply your equation with some lambda’s: the interpretation does not make sense if the lambdas are not chosen according to the Lagrangian theory.
- What is the univariate Gaussian baseline? Is this some equation in the paper, or some completely new thing? Can you give a definition? What is empirical estimator? What is standard estimator? What is OAS? Can you elaborate what you estimating, and what’s the story here? I think you are estimating something from the data, but not sure what. Maybe the precision structure? But why do you need the precision.. I thought that you needed the distribution of internal angles that are safe wrt too large absolute coordinate changes. Does this play a role in the experiments?
- What are local and global constraints? What is “valid” ramachandran distribution? What does “improved” 3D variance mean? What do you improve? What is “full covariance matrix”? Can you give a definition. Why is it full? How big is it? Covariance of what..? “that approximates distribution better…” Which distribution? Can you give a definition? What it the prior? Is it the eq 1? What is sigma_k,data? “underestimates the fluctuations” What fluctuations? Of whose? What is the “standard estimator”? What are you trying to achieve in the experiments? Can you clarify these points?
- How does the VAE impose global constraints? What are global constraints?
- Fig4: What is ramachandran reference? Where does the reference come from? What does it represent? What are the Rama samples? Where do they come from? Do these have something to do with data, or do they come from the model? What is MD reference? What does it represent or do?
- Fig5: What does this figure show? Where do these energy landscapes come from? What did you learn here? Did you just learn to match the MD reference panel with the VAE? What are the axes of this panel? I’m pretty lost what you did in the experiments, or what are you trying to demonstrate. The experiments do not discuss constraints or varying the angles in coordinate-safe ways at all, so I’m wondering if this is the goal anymore. Can you help me understand?
- What does “fitting density of structural ensemble” mean? What is an ensemble? Why is this an open problem? Are you talking about conformers?
- “We obtain samples that are guaranteed to fulfil physical constraints..”. With the amount of \approx in the paper, this is very hard to believe. Do you really guarantee **this? In what part of the paper is this guarantee made?

As you can see I lost track around halfway point when the story changes from safe angular variations to generating proteins. I don't see how these two stories connect. I'm looking forward to clarifications.

---

> ### Author Response · Authors · 2023-11-17
> **Response to reviewer U1Uk [1 of 2]**
>
> We thank reviewer U1Uk for their interest in our paper and their in-depth review. We have attempted to pool some of the concerns and questions together below and we hope to have addressed all points, but please let us know if something is missing. We look forward to further discussions.
>
> * **Central goal of the method and role of the VAE**
>
>   The central problem in this paper is density modelling, i.e. modelling protein fluctuations. We do density modelling in terms of internal coordinates, while imposing constraints in 3D space. These 3D constraints are global constraints which determine the overall shape of the protein, as opposed to local constraints which ensure e.g. valid bond lengths and no local clashes. We show how we can infer a covariance matrix over internal coordinates from a single mean structure using the Lagrange formalism. Subsequently, we train a reconstruction VAE on a collection of protein structures, also called a protein ensemble. A protein ensemble can be a collection of structures that exhibit small fluctuations (unimodal setting) or larger conformational changes (multimodal setting). The VAE decoder outputs a mean over internal coordinates, which is used to infer the covariance matrix corresponding to that mean, and together they parameterize a multivariate Gaussian distribution, from which we can draw samples. Through encoding and decoding angles (reconstruction), the decoder learns to map from $z$ to valid internal coordinate means, such that in generative mode, we can simply draw samples from the prior over $z$ and get valid output distributions. Through the latent variable, the VAE can capture multiple conformational modes by predicting different means and corresponding covariance matrices.
>
> * **Baselines**
>
>   We agree that the results are more meaningful with additional baselines, please see general response. We have also elaborated the explanation of the baselines to be more understandable.
>
> * **Predicting Lagrange multipliers**
>
>   First of all, we would like to emphasize that in our setting the constraints, $C$, are not user-defined, but are to be inferred by the model based on the input data. Having said that, we absolutely concur that in an ideal setting, one would solve for the Lagrange multipliers given the constraints, whether they are user-defined or predicted by a model. However, for the constraints we want to impose we cannot solve for $\lambda$ in closed form. Therefore we opted for predicting the Lagrange multipliers directly, which is essentially an indirect way of predicting $C$ (in other words, the lambdas are a proxy for $C$). To illustrate this, we have included a new appendix C.1 (see also general response), which shows the correspondence between constraints $C$ (calculated using Eq. 9) and atom fluctuations across VAE samples.
>
>   Moreover, one of the newly added baselines, "$\kappa$-prior (learned)", is a VAE trained to predict the variance directly, without including our constraints. This baseline does much worse compared to the full setup with constraints, which indicates that the constraints we incorporate are meaningful.
>
> * **Mathematical derivations**
>
>   We apologize for any lack of clarity and rigor regarding the mathematical derivations. As described in the general response, we have now elaborated the relevant sections (3.1-3.4) for improved understanding. We have incorporated all suggestions to the best of our ability, and would be happy to make further clarifications if necessary.
>
> * **Ramachandran distributions**
>
>   Ramachandran plots are a common way to visualize dihedral distributions by plotting the torsional angles around the $N-C_\alpha$ bonds ($\phi$) and $C_\alpha-C$ bonds ($\psi$) against each other (See section 4.2, "Unimodal"). The resulting distribution shows clusters for different secondary structural elements, e.g. beta sheets, right-handed helices, and left-handed helices. In our application, we can check qualitatively if the distribution is valid by comparing to the reference distribution (either simulation-based: MD, or experimental: NMR). We have updated Figure 4 to have more clear labels concerning the Ramachandran distributions.

---

> ### Author Response · Authors · 2023-11-17
> **Response to reviewer U1Uk [2 of 2]**
>
> This is our continued response to reviewer U1Uk.
>
> * **"Full" covariance matrix**
>
>   The full covariance matrix refers to the covariance over all dihedrals and bond angles (named $\boldsymbol{\kappa}$ in the paper), as opposed to a more standard setting of only considering the diagonal (i.e. the variance). For $M$ atoms, this corresponds to a matrix with $(2 \times M - 5)^2$ entries. The reason we emphasize the "fullness" of the covariance matrix is that we can infer it from predicting only M Lagrange multipliers.
>
> * **Prior**
>
>   We agree that the term "prior" can be a bit confusing, since there is also a prior for the latent space of the VAE. We now clearly separate this "$z$-prior" from the "$\kappa$-prior", where the latter is our prior belief over the distribution of $\Delta \kappa$, based on the variance over these internal coordinates in the training set. The reciprocal of this variance, makes up the diagonal of our prior precision matrix. This is shown in Figure 3 and explained in Section 3.1 (Setup) and Section 3.5 ("VAE model architecture").
>
> * **TICA explanation**
>
>   We have now elaborated our explanation for the TICA plots, please see general response.
>
> * **Density modelling: an open problem?**
>
>   Density modelling is an established and very active field of research, e.g. [1, 2, 3, 4] (as also mentioned in response to reviewer jsg1), and is not considered a solved problem.
>
> * **“We obtain samples that are guaranteed to fulfil physical constraints of the protein topology (e.g. bond lengths, and bond angles)”**
>
>   What we are saying in this specific sentence is that, by construction, the sampled structures will be guaranteed to adhere to a correct local protein topology, purely by choosing to model the protein in internal coordinate space. This means e.g. valid bond lengths and no local clashes. Chemical integrity is one of the main benefits of working with internal coordinate representations, as was also outlined in the second paragraph of Section 2.1. The approximations made in the paper only relate to global constraints. We have added the word ``local’’ to this sentence to make the distinction more clear.
>
> \
> *References*
>
> 1\) Noé, Frank, et al. "Boltzmann generators: Sampling equilibrium states of many-body systems with deep learning." Science 365.6457 (2019): eaaw1147.
>
> 2\) Zhong, Ellen D., et al. "CryoDRGN: reconstruction of heterogeneous cryo-EM structures using neural networks." Nature methods 18.2 (2021): 176-185.
>
> 3\) Ingraham, John, et al. "Illuminating protein space with a programmable generative model." BioRxiv (2022): 2022-12.
>
> 4\) Arts, Marloes, et al. "Two for one: Diffusion models and force fields for coarse-grained molecular dynamics." Journal of Chemical Theory and Computation 19.18 (2023): 6151-6159.

---

> > ### Comment · Reviewer_U1Uk · 2023-11-20
> > **resp**
> >
> > **On Lagrangian.** I’m still confused what are you actually doing. Are you doing the method of Lagrange multipliers (ie. https://en.wikipedia.org/wiki/Lagrange_multiplier), or something else? I don’t think you are since you should compute the stationary points of the lagrangian wrt all variables, and currently you are doing something else. I then wonder what is the “Lagrange formalism” you are talking about. Does the procedure you do in sec 3.0-3.4 have a name, or precedent in literature? Is this a known procedure or something you invented? Does it have something to do with the above Wikipedia page? I’m really struggling to find a common ground or basis of understanding.
> >
> > From my perspective you introduce a well-defined Lagrangian, and then apply some algebra to find some relationships between lambda’s and other variables. Ok, sure, but I don’t think this means in general anything (!). The constraints are only satisfied at stationary points of the Lagrangian, that is, when we have zero gradient wrt lambdas. I don’t think this is satisfied, so then I’m unsure what the result means, or if it has any meaning. In the end you don’t even do any optimisation wrt the Lagrangian, which is the premise of Lagrange method: we want to do constrained optimisation. So err.. I’m pretty lost. Can you clarify or help me understand?
> >
> > **Role of C.** You mention that C’s are not user defined. Err… what? My understanding is that C is the variance of the absolute coordinates when your angles follow some distribution. Surely you would then set this by hand to eg. 0.5 ångstroms (or something) and find an angle distribution that will give you this. But if the C’s are not user defined, then this intuition goes out the window. Can you help me understand what the C’s mean? It seems that there are also two different Cs, the C_m and C_n..
> >
> > **Guarantees.** Can you clarify what guarantees of physical constraints are you after in this paper? I thought the whole point of the paper was to obtain angle distributions such that you have no absolute coordinate clashes that come from compounding angles over the protein backbone. But in your response you say that you want to adhere to local topology. Ok, but isn’t this trivial by just predicting angles that are not too crazy? Surely pretty much any model is locally non-clashing (ie. the neighboring atoms don’t clash). Can you clarify?

---

> > > ### Author Response · Authors · 2023-11-20
> > >
> > > * **Lagrangian**
> > >
> > >   We apologize for not making this clear. It is indeed just the method of Lagrange multipliers we are using. Actually, it is very similar to the approach under the header "Example 4 (Entropy)" on the wikipedia page you mention (https://en.wikipedia.org/wiki/Lagrange_multiplier). In our case, it is a density rather than a discrete distribution, and its a relative entropy (i.e. KL divergence) rather the entropy, but otherwise the idea is the same. We write up the lagrangian, differentiate it and set to zero to find the stationary point. This "MaxEnt" technique for finding maximally non-informative distributions under a given set of constraints was pioneered by ET Jaynes in a paper in 1957 [1] for the discrete case, and later generalized to the continuous case in the 1960ies. A brief introduction is given on this wikipedia page: https://en.wikipedia.org/wiki/Principle_of_maximum_entropy.
> > >
> > >   As you can see on the page above (see "Continuous case"), solutions turn out to take an exponential form. Since we recognize the quadratic form in our exponent as a Gaussian, we don't have to solve for the normalization constant. All that is left to do is determine the value of the lagrange multipler lambda in terms of the constraint $C_m$. Normally, these $C_m$ constraints would be "user-defined" as you say, and it would be appropriate to find the corresponding lambda to realize the constraint on $C_m$. However, in our case, our network predicts the allowed level of fluctuations per atom (see comment below). Since we have a direct relationship between $C_m$ and the lagrange multiplier (eq 9 in our paper) we can therefore directly predict lambdas instead of the $C_m$ values. Doing so implicitly sets the fluctuation level for each atom.
> > >
> > >
> > > * **Role of $C$**
> > >
> > >   Your intuition is correct - you can think of it as choosing an expected fluctuation level $C_m$ for each atom in the chain. Rather than saying that these values are "user defined", we just phrase it in terms of "fitting" these fluctuations such that they best fit the the target density. In other words, the model fits the level of fluctuation for each individual atom such that it corresponds best to the distribution of the training data.
> > >
> > >   One additional complication is that since the Lagrange formalism above is employed as part of the decoder in a variational autoencoder, all "predictions" we discussed above are conditioned on a latent variable $z$. Different choices of $z$ will correspond to different modes of the distribution we are trying to fit, and some will require large fluctuations than others. We thus have $C_m$ values for each atom $m$, but these are dependent on $z$.
> > >
> > >   Thank you for pointing out the inconsistency for the index variable ("$m$" to "$n$" in eq 5), we agree that this can be confusing.  They are the same entity. We apologize and have now updated the manuscript correspondingly (the index is now $m$ everywhere).
> > >
> > >
> > > * **Guarantees**
> > >
> > >   Sorry for not making this sufficiently clear. Our goal is to fit the density of a distribution of protein structures. First, we make the choice of working in internal coordinates. We motivate this choice by the fact that internal coordinates ensure that local geometric constraints (bond lengths, bond angles between subsequent atoms) are fulfilled *by construction*. This would not be the case if we had tried to model the density directly in Cartesian coordinates.
> > >
> > >   Once we have made the choice to work in internal coordinates, our biggest challenge becomes exactly as you state: "absolute coordinate clashes that come from compounding angles over the protein backbone". Our method is indeed designed to solve exactly this problem (there are no local geometry issues at all *because* we choose to work in internal coordinates).
> > >
> > > \
> > > *Reference*
> > > 1) Jaynes, Edwin T. "Information theory and statistical mechanics." *Physical review* 106.4 (1957): 620.

---

> > > > ### Comment · Reviewer_U1Uk · 2023-11-21
> > > > **resp**
> > > >
> > > > Ok, now the Lagrangian stuff makes sense. I'll raise my score.
> > > >
> > > > In future I would recommend the author's to try to frame the work more around ML objectives. I believe the work would be more convincing if you would incorporate some (eg. reconstruction) benchmarks, and perhaps apply the work towards conformer generation which seems to be an ideal usecase for this.

---

> > > > > ### Author Response · Authors · 2023-11-21
> > > > >
> > > > > We sincerely thank the reviewer for raising their score, and for all their constructive suggestions that helped us improve the manuscript.
> > > > >
> > > > > The reported results on fluctuations along the atom chain (Figure 4) and TICA landscapes (Figure 5) are the result of sampling by our VAE (see also appendix A for experimental details and appendix C.2 and C.3 for sample visualizations), which generates conformers based on the learned density. In the current version of the manuscript, we also compare the conformer-generating performance against several baselines, including a flow-based model [1] which is, to the best of our knowledge, state-of-the-art for internal-coordinate density modelling (see general response).
> > > > >
> > > > > \
> > > > > *Reference*
> > > > > 1) Köhler, Jonas, et al. "Flow-matching: Efficient coarse-graining of molecular dynamics without forces." *Journal of Chemical Theory and Computation* 19.3 (2023): 942-952.

---

### Official Review · Reviewer_jsg1 · 2023-10-31

**Soundness:** 3 good
**Presentation:** 3 good
**Contribution:** 3 good
**Rating:** 6
**Confidence:** 3

**Summary:**

The authors parameterize protein structure with internal coordinates and estimate those quantities under the framework of a multivariate gaussian distribution. The authors then train a neural network to simulation data to estimate these parameters, and then sample from it, comparing to the simulation they trained on.

**Strengths:**

I think it is great the authors are able to define the limitations of their work with respect to the amount of data available for a protein of interest.

The description of number of parameters to be estimated (2.1) is very clear and helpful for intuition of the model components.

I really like the “Variance along the atom chain” figures. I think they are really clear and show where the “wiggles” happen.

**Weaknesses:**

I still have difficulty with the motivation of using ML models trained on simulation data. Why not just run the simulation then? What does this improve beyond the simulation framework?

I would like the TIC plots in Figure 5 better explained. Also, why does it seem that there are many points cut off from the figure?

It is odd to me that there are no scalar metrics in this work, especially those that allow comparison to other methods.

While I understand the potential for this approach, I feel like the authors could do a better job of describing how this method could actually solve understanding of potential protein structure fluctuations.

**Questions:**

“To ensure this, we add an auxiliary regularizing loss in the form of a mean absolute error over λ −1 ,” Does this not introduce a Laplace likelihood (or prior) into the posterior? It’d be great to explore that relationship and decision a bit more.

In Figure 4, looking at the Ramachandran plots, it seems like the samples are not capturing the actual variance in the data. Why is this? Is a Gaussian prior the right prior? What about Laplace or StudentT? Are there other parameters that could fully capture this variability, assuming that it is real?

I would be curious of proteins that have very large degrees of freedom, or move a lot during function.

---

> ### Author Response · Authors · 2023-11-17
> **Response to reviewer jsg1**
>
> We sincerely thank reviewer jsg1 for their interesting questions and helpful suggestions. We will answer in detail below.
>
> * **Density modelling (ML models trained per system)**
>
>   In this paper, we are training one model per protein using simulation data or experimental data to model the density over internal coordinates. Density modelling is an established and active field of research, e.g. [1, 2, 3, 4], with direct and indirect applications. Capturing a distribution from which we can sample cheaply (which is enhanced by the fact we model the backbone only) is useful for data augmentation, especially in a low data setting. Moreover, density modelling per protein can be seen as a step towards modelling complexes, doing cheap simulations, analyzing function and, ultimately, generalize across proteins.
>
> * **TIC plots and quantitative metrics**
>
>   We thank the reviewer for the great suggestions, and refer to the general response for our answer.
>
> * **How could this method help us understand protein fluctuations?**
>
>   This is a very interesting point, and we would argue that our method works in the opposite direction. We incorporate our knowledge of protein fluctuations as an inductive bias in the proposed method, thereby creating a more meaningful way to incorporate constraints.
>
> * **Regularizer**
>
>   We agree that our auxiliary loss corresponds to multiplying with a Laplace prior (L1 loss). We chose this auxiliary loss because our approximation only holds for smaller fluctuations. Without the regularizer,the  global constraints can become too weak, and the covariance matrix can become invalid. The hyperparameters $a$ and $w_{\mathrm{aux}}$ can be used for tuning (see appendix D).
>
> * **Figure 4: capturing the actual variance**
>
>   We agree that our method does not always capture the true variance of the data. We believe this to be partly due to hyperparameter choice, the first-order approximation for fluctuations (Eq. 7) and, indeed, the Gaussian assumption. However, we stress that modelling a complex protein like 1pga in a limited data regime is highly challenging, and it might be that our mean encoder and decoder are not expressive enough in this setting. One possible solution could be introducing a hierarchical VAE to model larger local fluctuations, but this is beyond the scope of the current paper.
>
> * **Proteins with large degrees of freedom**
>
>   We believe our multimodal test cases to exhibit large degrees of freedom, as folding and unfolding involves dramatic conformational changes. However, we would like to emphasize that our general-purpose method is designed for improved modelling of smaller fluctuations, and that large degrees of freedom can be realized through model architecture choices. For example, our proof-of-concept VAE realizes large conformational changes through different predicted means (and corresponding covariance matrices).
>
> \
> *References*
>
> 1\) Noé, Frank, et al. "Boltzmann generators: Sampling equilibrium states of many-body systems with deep learning." *Science* 365.6457 (2019): eaaw1147.
>
> 2\) Zhong, Ellen D., et al. "CryoDRGN: reconstruction of heterogeneous cryo-EM structures using neural networks." *Nature methods* 18.2 (2021): 176-185.
>
> 3\) Ingraham, John, et al. "Illuminating protein space with a programmable generative model." *BioRxiv* (2022): 2022-12.
>
> 4\) Arts, Marloes, et al. "Two for one: Diffusion models and force fields for coarse-grained molecular dynamics." *Journal of Chemical Theory and Computation* 19.18 (2023): 6151-6159.

---

### Official Review · Reviewer_QUDF · 2023-11-01

**Soundness:** 3 good
**Presentation:** 3 good
**Contribution:** 2 fair
**Rating:** 5
**Confidence:** 3

**Summary:**

This paper investigates the internal-coordinate density modeling of protein structures. One issue in internal-coordinate modeling is: small fluctuations in internal coordinates can lead to large fluctuations of atoms remotely.
  - The authors first conduct some simulations in section 2.2 to show that even while sampling from the true mean/variance of internal-coordinates, the global atom positions fluctuate a lot.
  - The authors then proposed adding a constraint on the variance of atom positions, and finding another multivariate Gaussian model on internal-coordinates that is closest (in terms of KL divergence) to the given distribution, while satisfying the variance constraints. This is realized by an approximate solution to Lagrangian formalism.
  - The overall framework is VAE, with the above-mentioned correction to the Gaussian distribution.
  - Experiments are conducted mostly on MD simulation data in two scenarios: where the structures have small fluctuations and large fluctuations.

**Strengths:**

- This paper tackles a well-motivated issue in internal-coordinate modeling.
  - The proposed solution makes sense under the framework of multivariate Gaussian model.  - The experiments do not include enough baselines to validate the practical usefulness of this method. Many related DL methods are mentioned in related work, but they are not compared in the experiments. Although the authors mentioned the main focus of this work is internal-coordinate modeling, I am still interested in a more thorough comparison.

**Weaknesses:**

- The experiments do not include enough baselines to validate the practical usefulness of this method. Many related DL methods are mentioned in related work, but they are not compared in the experiments. Although the authors mentioned the main focus of this work is internal-coordinate modeling, I am still interested in a more thorough comparison.

**Questions:**

1. Section 3.5: U-net is used to estimate the values for the Lagrange multipliers \lambda for each constraint. Is there any supervision or auxiliary loss on \lambda?
  2. Restricting the fluctuation will restrict the representation power of the model. How to balance that except for hand-tuning the hyper-parameters?
  3. Any analysis to compare the models learned on the unimodal scenario and multimodal scenario?
  4. Can the model generalize to generate conformers for new proteins?

---

> ### Author Response · Authors · 2023-11-17
> **Response to reviewer QUDF**
>
> First of all, we would like to thank reviewer QUDF for their insightful comments, which we will address point-by-point below.
>
> * **Baselines**:
>
>   We agree that adding more baselines is very valuable. Please refer to our general response for the newly included baselines.
>
> * **Auxiliary loss on $\lambda$**
>
>   It is indeed important to have a regularizer on $\lambda$ values, which we employ in the form of an MAE auxiliary loss (see Section 3.5).
>
> * **Balancing the representation power of the model**
>
>   In the current setup, balancing the representation power of the model is mostly done through hyperparameter tuning (see also appendix D). Through the latent variable model (VAE), we gain some flexibility by predicting different means (and corresponding covariance matrices). However, in a low data regime and especially for more complex proteins like 1pga, it is still non-trivial to do perfect density modelling regardless of the chosen architecture.
>
> * **Unimodal scenario vs multimodal scenario**
>
>   We stress that our main contribution is to model small fluctuations correctly, which is also where we see the most distinct difference between our method and the baselines. We included the multimodal scenario to show how our general-purpose method can potentially be used to model larger conformational changes, which is mostly realized through the latent variable mapping to different means.
>
> * **Generalization across proteins**
>
>   Generalization to new systems is currently beyond the scope of the paper, as this would require a large and diverse dataset, which we don’t have at our disposal. However, we suggest that our method for incorporating constraints in internal-coordinate modelling could facilitate the next steps towards generalization when incorporated in a highly expressive model that is trained on sufficient amounts of data.

---

### Author Response · Authors · 2023-11-17
**General response**

Response to all reviewers, ACs, and readers:

We would like to thank the reviewers for their valuable and constructive feedback.

As a global response, we will summarize the major changes we have included in the newly uploaded manuscript:

* **Baselines** [QUDF, U1Uk]

  We now include a total of 4 different baselines in the manuscript:
  1. "$\kappa$-prior (fixed)": a VAE trained to predict $\mu_\kappa$ given a fixed covariance matrix that is equal to $\boldsymbol{\Sigma}^{-1}_{\rm{\boldsymbol{\kappa}, prior}}$. In other words, this is the same as our full VAE setup, but omitting the imposed 3D constraints.
  2. "$\kappa$-prior (learned)": a more standard VAE-setting where the decoder directly outputs a mean and a variance (i.e.\ a diagonal covariance matrix).
  3. "Standard estimator": this baseline does not involve a VAE, but samples structures from a multivariate Gaussian with a mean based on the dataset and a precision matrix computed by a standard estimator.
  4. "Flow"[1]: a flow-based model, which, to the best of our knowledge, is the current state of the art in density modelling for internal coordinate representations.

  We have updated the main figures to include the new baselines, and also show sample visualizations for all baselines in appendix C.2.

* **TICA explanation** [jsg1, U1Uk]

  We have elaborated the explanation about the TICA free energy landscapes in Section 4.2, making this metric more intuitive.

* **Quantitative results** [jsg1]

  We have added quantitative results, corresponding to Figure 4 and Figure 5, as appendix B. For the atom fluctuations, we report the MSE to the reference. For the TICA landscapes, we report the Jensen-Shannon distance to the reference free energy landscape.

* **Additional test case and improved clarity for Figure 4** [U1Uk]

  We have included an additional protein for the unimodal setting in a low data regime: 1fsd. This is an NMR dataset with a more mixed secondary structure compared to the other NMR dataset (1unc). Moreover, we have reorganized Figure 4 to be more clear and informative.

* **Improved clarity and rigor in mathematical derivations** [U1Uk]

  We have extended Sections 3.1-3.4 to include more intermediate steps, and added footnotes and references where necessary for enhanced comprehensibility.

* **Evaluating constraints** [U1Uk]

  In the new appendix C.1, we added an evaluation of the constraints that were calculated given a set of predicted Lagrange multipliers (Eq. 9), and compare the result to the atom fluctuations of VAE samples, demonstrating that these correlate quite well.


Apart from this global response, we will answer the questions and concerns of each reviewer in more detail below. We have addressed all concerns and suggestions to the best of our ability, and we believe that this has improved our manuscript in terms of clarity, rigor, and significance. Of course, we are open to further discussion, and are happy to give additional clarifications.

\
*Reference*

1\) Köhler, Jonas, et al. "Flow-matching: Efficient coarse-graining of molecular dynamics without forces." *Journal of Chemical Theory and Computation* 19.3 (2023): 942-952.

---

> ### Author Response · Authors · 2023-11-21
>
> To all reviewers, ACs, and readers,
>
> \
> We thank the reviewers for their comments, which we have meticulously addressed in the current version of the manuscript, as outlined in the general response above. The main new contributions feature: 1) more baselines, including a flow-based model that is state-of-the-art in internal-coordinate density modelling, 2) a new test case (protein 1fsd, NMR dataset), 3) quantitative results, and 4) extended explanations for mathematical derivations and TIC analysis.
>
> We highly value your feedback, and would be happy to engage in further discussions and provide clarifications where necessary.

---

### Author Response · Authors · 2023-11-23

To all reviewers, ACs, and readers,

As we are approaching the end of the discussion period, we hope all reviewers got a chance to look at the changes we have made to the manuscript in order to address all their concerns. We believe these changes, mostly inspired by suggestions from the reviewers, have significantly improved the paper. Moreover, we have provided clarifications and discussions as a response to each individual reviewer. We hope the reviewers will take our efforts into consideration as the process enters the next phase.

---

### Meta-Review · Area_Chair_UiAZ · 2023-12-06

**Metareview:**

Ultimately, I think this paper would benefit from an additional round of revision and new reviews after addressing reviewer concerns. Two reviewers commented, and I tend to agree after reading myself, that the presentation quality and clarity here could be significantly improved. A lot of space in the paper is devoted to relatively trivial equations or quantities (e.g., the KL divergence in eqn. 2) without explicitly defining key derivation steps, and I had many of the same initial questions as Reviewer U1Uk. While the authors were able to clarify many points of the paper in an extremely extensive author feedback, I just think that the final impact on the final writing of the paper here would be large enough that it's difficult to picture the paper in its final form. Furthermore, beyond writing improvements, the authors are including additional experiments and baselines (e.g., the general response). There's just enough editing being promised here that the paper should be reviewed again.

**Justification For Why Not Higher Score:**

The changes to both the writing of the paper and the experimental results seem fairly substantial, and I'm essentially placing this in a "revise and resubmit after major revisions" kind of category.

**Justification For Why Not Lower Score:**

N/A

---

### Decision · Program_Chairs · 2024-01-16

Reject